# Effects of Nitrogen Fertilizer on Photosynthetic Characteristics and Yield

Hafeez Noor [ID], Pengcheng Ding, Aixia Ren, Min Sun * and Zhiqiang Gao

College of Agriculture, Shanxi Agriculture University, Jinzhong 030801, China; hafeeznoorbaloch@gmail.com (H.N.)
* Correspondence: sunmin@sxau.edu.cn

**Abstract:** This study aims to understand the influence of chlorophyll fluorescence parameters on the yield of winter wheat in some areas of China. Nitrogen (N) application is believed to improve photosynthesis in flag leaves, ultimately increasing the final yield. The results from different parameters of research showed that the grain number per spike improved by 12.2% and the 1000-grain weight by 7.3%, respectively. At 20–30 days after anthesis, the activities of superoxide dismutase (SOD), peroxidase (POD), and soluble protein in flag leaves of N150 were found to be the most effective. The grain yield under N manure partial substitution for N fertilizer treatment increased by 23 and 15%, respectively. The important implications of photosynthetic characteristics of variable fluorescence yield of the light-adapted state are that the contents of clear, ball, alcohol soluble, gluten, protein, and flour quality showed an increasing trend, while the contents of amylose, amylopectin, total starch, and direct/branch ratio were decreased of wheat. The net photosynthetic rate, transpiration rate, and relative chlorophyll content of wheat. The outcome of the present investigation suggests that chlorophyll fluorescence parameters could be a valuable insight into understanding yield stability under stress conditions. Moreover, the investigated parameters could be useful criteria for the selection of photosynthetic energy under varying nitrogen application levels.

**Keywords:** chlorophyll fluorescence; grain protein; starch contents; yield component; photosynthetic characteristic; anthesis; wheat harvest





## 1. Introduction

Rainfed agriculture in Asian and Pacific regions accounts for approximately 70% of the region's arable land, and 60–80% of the global food supply comes from these rainfed lands [1]. The successful cultivation and promotion of superior varieties are the basis for the continuous increase in wheat yield and grain quality [2]. Under drought conditions, the plant stomata rates closed, due to which the plant's absorption of $CO_2$ reduces, and the rate of photosynthesis, growth, and yield of the plant is affected badly [3]. High planting density has been adopted in sprout production systems, which improved photosynthesis, and influenced plant height, architecture, and synthesis of chlorophyll. Seeds consume a lot of energy and resources during germination. High amounts of antioxidant substances are synthesized during germination [4]. Leaf senescence comprises a series of biochemical and physiological events from the fully expanded state until death. The leaf duration after full expansion depends strongly on the water conditions and crop species; some researchers have reported that the post-anthesis senescence in cereals affects the whole plant, with organs closest to the developing grains flag leaves generally senescing last [5]. Within a certain range, with the increase in nitrogen application level, the protein content of wheat grains increases, but excessive or insufficient nitrogen application will reduce the transport of accumulated nitrogen before anthesis to grains, and affect the protein content of grains [6]. The photosynthetic duration is closely related to leaf aging. The premature photosynthetic nitrogen transport in wheat results in premature canopy leaf senescence, and grain yield reduction [7,8]. The green leaf area has effects on the grain yield, and shaded leaves become

senescent earlier compared to unshaded leaves [9]. Nevertheless, increasing the N content, the Net photosynthetic rate ($P_N$), stomatal conductance ($gs$), transpiration rate ($Tr$), and Ci intercellular $CO_2$ concentration, as well as increasing the substomatal $CO_2$ concentration ($C_i$), are considered as the value of wheat stomatal restriction in different plant populations. It was reported that N also improved yield and components in wheat [10]. However, the process of nitrogen transport from wheat organs to grains is extremely complex in physiology. Field-scale studies have no way to determine which photosynthetic nitrogen and stored nitrogen transport starts first, or even synchronously, but the transport ratio of photosynthetic nitrogen and stored nitrogen can be calculated [11]. The effects on the net photosynthetic rate ($P_N$) vary in seed types and at different growth stages. The current scenario of climate change such as high temperatures requires the development of wheat with improved leaf gas exchange parameters. Wheat $P_N$ is not affected by high intensity of light, temperature, and low humidity [12]. The root-cutting maintained higher photosynthesis but significantly reduced transpiration and stomatal conductance [13,14]. Maintaining higher photosynthesis and reducing transpiration water loss are important for improving water use efficiency. The grain-filling process mainly depends on three aspects of material supply. The direct transport of photosynthetic from leaves and stem sheaths to grains during grain filling [15]. Therefore, dry matter accumulation and transport during the wheat grain-filling stage are very important processes for yield formation and water-use efficiency. However, there was little information on dry matter accumulation and transport during the wheat-filling stage, the effects of fertilizer on wheat yield, and water-use efficiency. In general, the N application of chemical fertilizers can increase biomass accumulation and thus yield. However, excessive biomass accumulation leads to excessive canopy leaf area, resulting in an excessive increase in transpiration and water loss, and ultimately excessive consumption of soil water [16]. In the long run, increased fertilizer application cannot continuously increase crop yield and water-use efficiency. N fertilizer application can significantly increase soil water content [17]. The efficient nitrogen uptake by matched crops improves crop yield and water-use efficiency [5,17]. N fertilizer utilization not only improved soil water retention capacity by improving soil aggregate components (>0.25 mm), but also significantly increased soil nutrient contents in the growth period, and ultimately significantly increased crop yield and water-use efficiency [18]. Nitrogen fertilizer is a major component of proteins and nucleic acids, and its application regulates plant growth [19]. Nitrogen fertilizer inhibits the efficiency of photosynthesis and irradiation to reduce grain yield, while the optimal concentration of nitrogen application can increase grain yield [20]. A previous study suggested that N fertilizer has a positive correlation with photosynthetic efficiency [21]. It mainly functions in phytohormone metabolism, chlorophyll degradation, nucleic acid degradation, protein degradation, nitrogen metabolism, lipid metabolism, anti-oxidation, enzymes related to aging to senescence, and transcription factor [22]. The objectives of this study were to understand (1) the effects of fertilizer on wheat yield, water-use efficiency, and photosynthetic characteristics; (2) the effects of N fertilizer on dry matter reuse during grouting stage; and (3) to study the response of physiological characteristics of nitrogen use, light use, and leaf senescence, yield formation, and quality of wheat.

## 2. Materials and Methods

The experiment was conducted in the wheat experimental base of Shanxi agricultural university in Taigu, Shanxi Province, China (E112°34′ E, N37°25′ N) from 2019 to 2022, which belongs to the temperate continental climate zone, with an average annual temperature of 10.4 °C. The greenhouse had concrete walls with a thickness of 20 cm, and an insulation layer of 10 cm was added to the outer walls; see Figure 1 and Table 1.

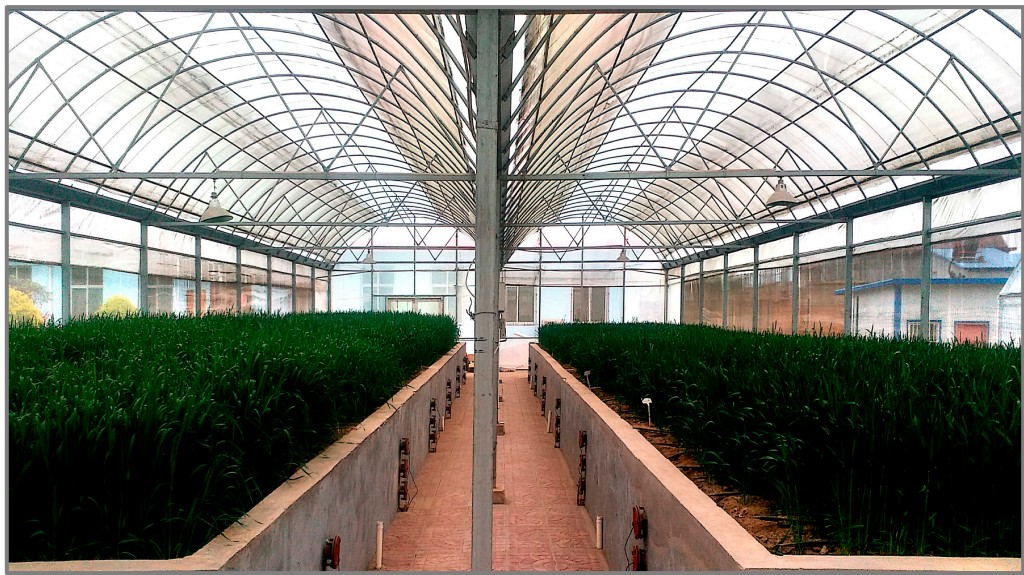

**Figure 1.** Greenhouse at experimental site of Shanxi Agricultural University Taigu Shanxi Province, China.

**Table 1.** Soil basic fertility of 0–20 cm soil layer at the experimental site in Taigu.

| Year | Organic Matter (g kg$^{-1}$) | Total N (g kg$^{-1}$) | Alkaline N (mg kg$^{-1}$) | Olsen P (mg kg$^{-1}$) | Available K (mg kg$^{-1}$) |
|---|---|---|---|---|---|
| 2019–2020 | 9.89 | 1.07 | 42.78 | 11.66 | 206.74 |
| 2020–2021 | 9.69 | 0.99 | 43.06 | 10.95 | 194.53 |
| 2021–2022 | 9.61 | 1.06 | 44.07 | 10.71 | 188.87 |

## 2.1. Experimental Design and Crop Management

The experiment was arranged in a split-plot design with the four N application rates: 0 (N0), 120 (N120), 150 (N150), and 210 (N210) kg ha$^{-1}$. A widely planted wheat N application rate of 97.5 kg ha$^{-1}$ in early October of each year. The plot size was 2 m $\times$ 4 m = 8 m$^2$. Before sowing, 150 kg P$_2$O$_5$ ha$^{-1}$ and 150 kg K$_2$O ha$^{-1}$ were evenly applied to the plots, respectively. The nitrogen fertilizer was applied to the base fertilizer. The seeds were sown on 9th November, planted in a manual with row spacing of 20 cm, and a sowing quantity of 225 kg ha$^{-1}$. Irrigation was applied using a drip irrigation system, with the application of 50 mm each time as measured with a water meter. All plants were harvested on 2 June. During the whole growing season, weed was well controlled by hand. Based on our previous farmers' survey (sample size, $n$ = 420), the smallholder farmers in the area apply fertilizers once before sowing.

### 2.1.1. Photosynthetic Pigment and Chlorophyll Fluorescence Parameters

The single-photon avalanche diode SPAD-502 measurements were conducted in the field between 10:00 and 16:00 h. The adaxial side of the leaves was always placed toward the emitting window of the instrument, and major veins were avoided. The area of the punched wheat leaf material was determined using an area meter (Minolta, Osaka, Japan) while two circular 1.0 cm diameter leaf discs from one side on the midrib were punched for wheat. The SPAD chlorophyll analyzer (Minolta, Osaka, Japan) was used to determine flag leaves at 0−5−10−15 days after anthesis. The SPAD value of the non-damaging method was adopted for the determination. The values were read from the middle position of the blades and measured 3 for each treatment.

### 2.1.2. Yield Components

At maturity, ten plants from each plot were randomly sampled from the inner rows for the determination of yield components such as ear number, seed number per ear, and

weight of thousand seeds. Plot grain yield was determined by harvesting all plants in the area of 20 m$^2$, shelled using a machine, and the grain was air-dried for the determination of grain yield.

### 2.1.3. Grain Total Sucrose, Soluble Sugar, and Starch Content

After the flowering period listed growth was consistent and on the same day flowering of wheat spike and peeling grain was placed in the oven and dried at 105 °C for 20 min and then at 75 °C for 12 h for dry weight. The quality and speed of weighing samples greatly affect the overall quality of the test. Then, the grain was weighed, and the phenol method was used to determine the content of sucrose and the ketone color method was used to determine the total soluble total sugar content, while $H_2SO_4$-$H_2O_2$-Phenol blue color method was used to determine the seed protein and component content [23].

### 2.1.4. Grain Protein and Its Components

Continuous extraction method was used to determine the protein components of grains, and the nitrogen content × 5.7 was the protein content and we repeated three times for each sample. The samples were dried at 105 °C for 30 min and then weighed at 75 °C. The seeds at maturity were naturally air-dried without drying. After grinding, nitrogen content of plant samples was determined by $H_2SO_4$-$H_2O_2$-indiophenol blue colorimetric method.

### 2.1.5. Physiological and Biochemical Traits Measurement

The contents of malondialdehyde (MDA) and osmotic solutes in fresh leaves were analyzed at the end of the drought stress period (at 5, 10, 15, 20, 25, and 30 days after anthesis). For preparing the crude extracts, 0.5 g of fresh leaves from three replicates in each treatment were homogenized in a pre-chilled mortar and pestle using 10 mL of 50 mmol L$^{-1}$ phosphate buffer (pH 7.8) containing 1% of soluble polyvinyl pyrrolidone (PVP). The homogenates were then centrifuged at $10,000 \times g$ for 20 min at 4 °C, and the resulting supernatants were used for enzyme activity and biochemical assays. The content of MDA was determined using 2-thiobarbituric acid (TBA) [22]. Briefly, 2 mL of extract solution was added to 2 mL of 0.5% TBA (dissolved with 15% trichloroacetic acid) and incubated in a water bath at 95 °C for 30 min. After centrifugation ($10,000 \times g$ for 10 min), the MDA content was determined as mmol g$^{-1}$ FW (fresh weight) by measuring the absorbance at wavelengths of 450 nm, 532 nm, and 600 nm using a spectrophotometer.

### 2.1.6. Statistical Analysis

The data of photosynthesis physiological and winter wheat growth yield was processed and statistically analyzed through Microsoft Excel 2010 and Sigma Plot 14.0 software to process data and draw graphs, and DPS7.5 was used for statistical analysis. A two-way ANOVA was used to study the main influence and interaction of variable fluorescence types on yield. When there was a significant interaction effect between photosynthesis physiological, SPAD, and yield, the least significant difference (LSD) method was used for variance analysis and independent T-test, and the significance level was set to $\alpha = 0.05$; differences were considered statistically significant when $p \leq 0.05$.

## 3. Results

### 3.1. Activities of Enzymes

At 0–15 days after anthesis, the activities of superoxide dismutase activities (SOD), peroxidase (POD), and soluble protein in flag leaves of N0 and N120 had no significant differences with those of N150 (Figure 2). At 20–30 days after anthesis, the activities of superoxide dismutase (SOD), peroxidase (POD), and soluble protein in flag leaves of N150 were significantly higher than those of N210, respectively. At 0–10 days after anthesis, the catalase (CAT) in flag leaves of N0 and N120 wheat showed no significant difference, but at 15–30 days after anthesis, the catalase (CAT) in N150 was significant. N150 the

superoxide dismutase (SOD) activity, peroxidase (POD) activity table, soluble protein and catalase (CAT) in flag leaves of wheat increased first and then decreased. In the two wheat growing seasons, malondialdehyde (MDA) in the flag leaf of wheat after anthesis increased and then decreased with the progress of grouting. At 0–15 days after anthesis, there was no significant difference in malondialdehyde (MDA) in flag leaves of N0 wheat. At 20–30 days after anthesis, the POD activity of N150 was significantly lower than that of 210. The malondialdehyde (MDA) of wheat flag leaves decreased with the increase in nitrogen application rate. The activities of superoxide dismutase (SOD) and peroxidase (POD) and soluble protein and catalase (CAT) in the kernel flag leaf of wheat after anthesis first increased and then decreased with the progress of grout.

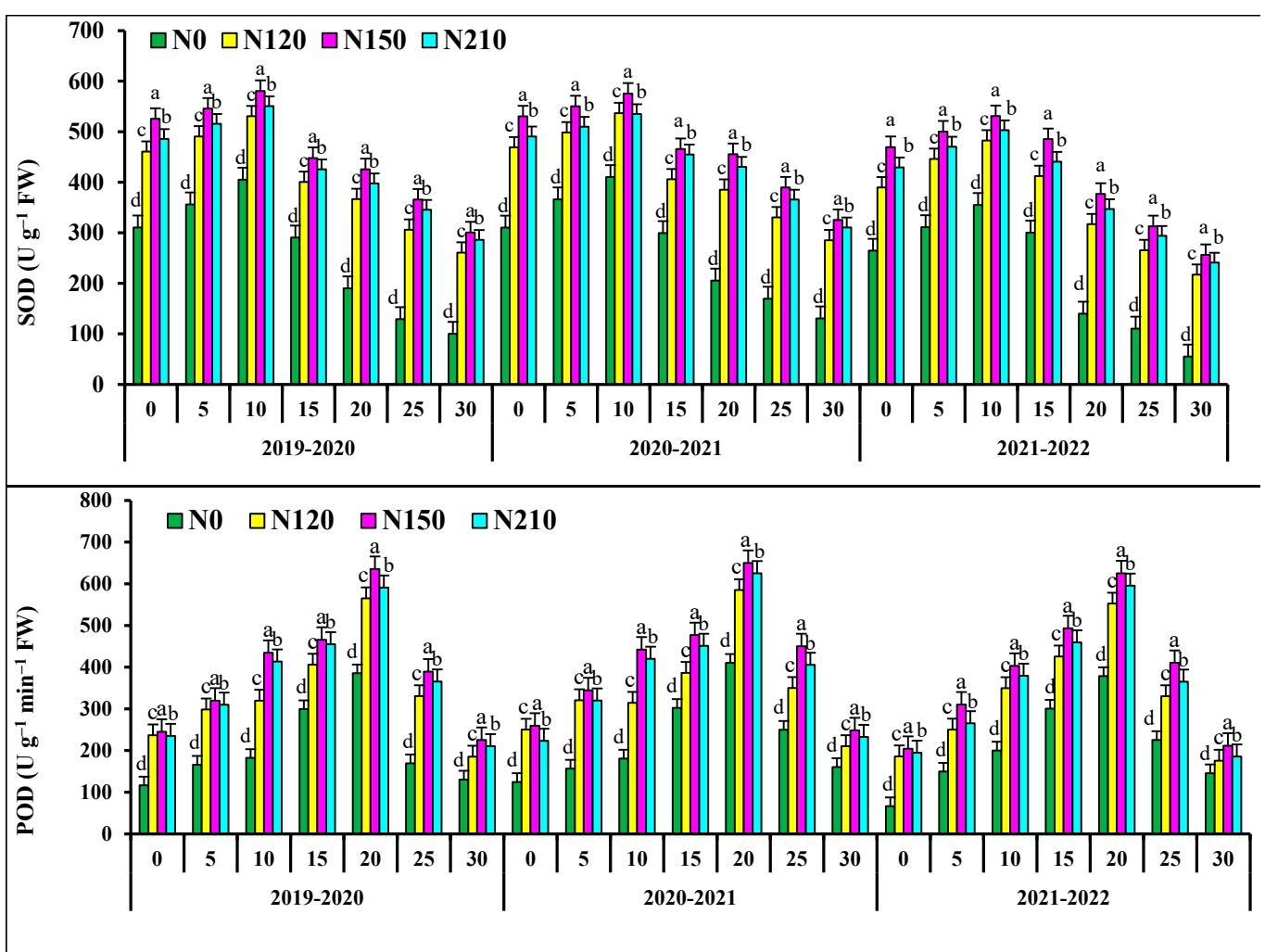

**Figure 2.** *Cont.*

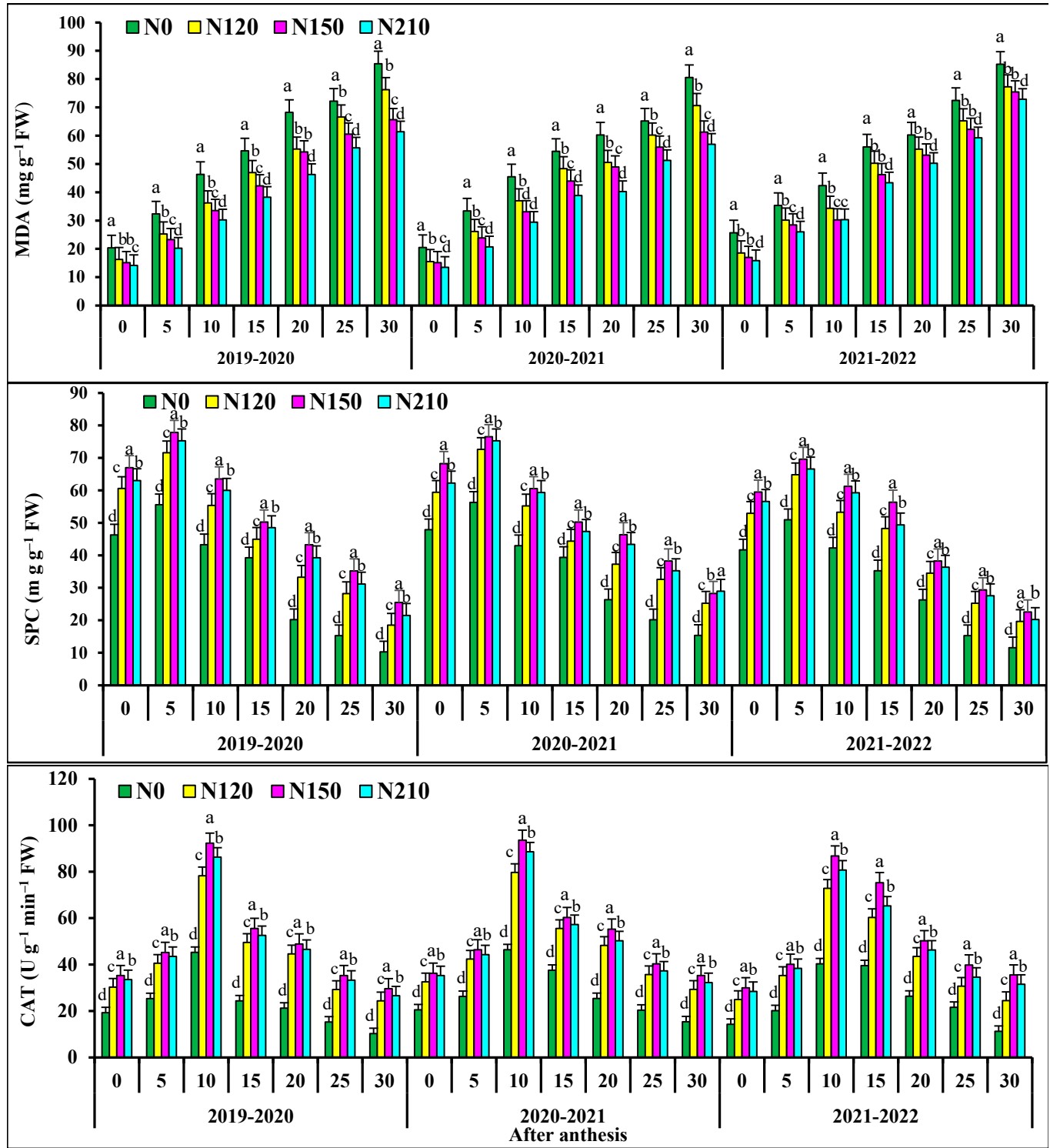

**Figure 2.** Effects of nitrogen application on fluorescence characteristic superoxide dismutase activities (SOD), peroxidase isozyme (POD), malondialdehyde concentration (MDA), soluble protein concentration (SPC), catalase (CAT), in wheat flag leaves after anthesis. Different lower cases indicate significant differences among treatments.

*3.2. Photosynthetic Characteristics*

Net photosynthesis $P_N$ reached the maximum value in the days after anthesis and began to decline after the flowering stage (Figure 3A). The maximum $P_N$ value of 20–30 d was 20.06, which was increased by 15.37%, 6.97%, and 5.88% compared to of nitrogen

rate, respectively, indicating that nitrogen application could significantly improve the photosynthetic rate of wheat leaves, among which medium nitrogen treatment had the most significant promoting effect. However, the high nitrogen application rate did not improve the net photosynthetic rate of wheat leaves but inhibited it to some extent. The appropriate demographic structure was useful for accumulating net photosynthetic rates, and N150 significantly enhanced the $P_N$ increase in winter wheat flag leaves, whereas excessive nitrogen use had particular inhibitory effects. There was a significant difference between them in N0 and N120 and a significant difference between the treatments in the after-anthesis.

Effects of different N rate comparisons of stomatal conductance gs are shown in Figure 3B. The 20–30 day, N150 was the highest, reaching 0.45 followed by 20–30 day, N210, and 10–20 day N150. The gs value was highest in the N150 compared to other treatments, and there was a significant difference between N0 and N120 treatments. There was a significant difference between treatments N120 and N150. However, there was a significant difference between N210. N150 had very significant effects on gs at each growth stage, but only N150 had a very significant interaction in the days after anthesis.

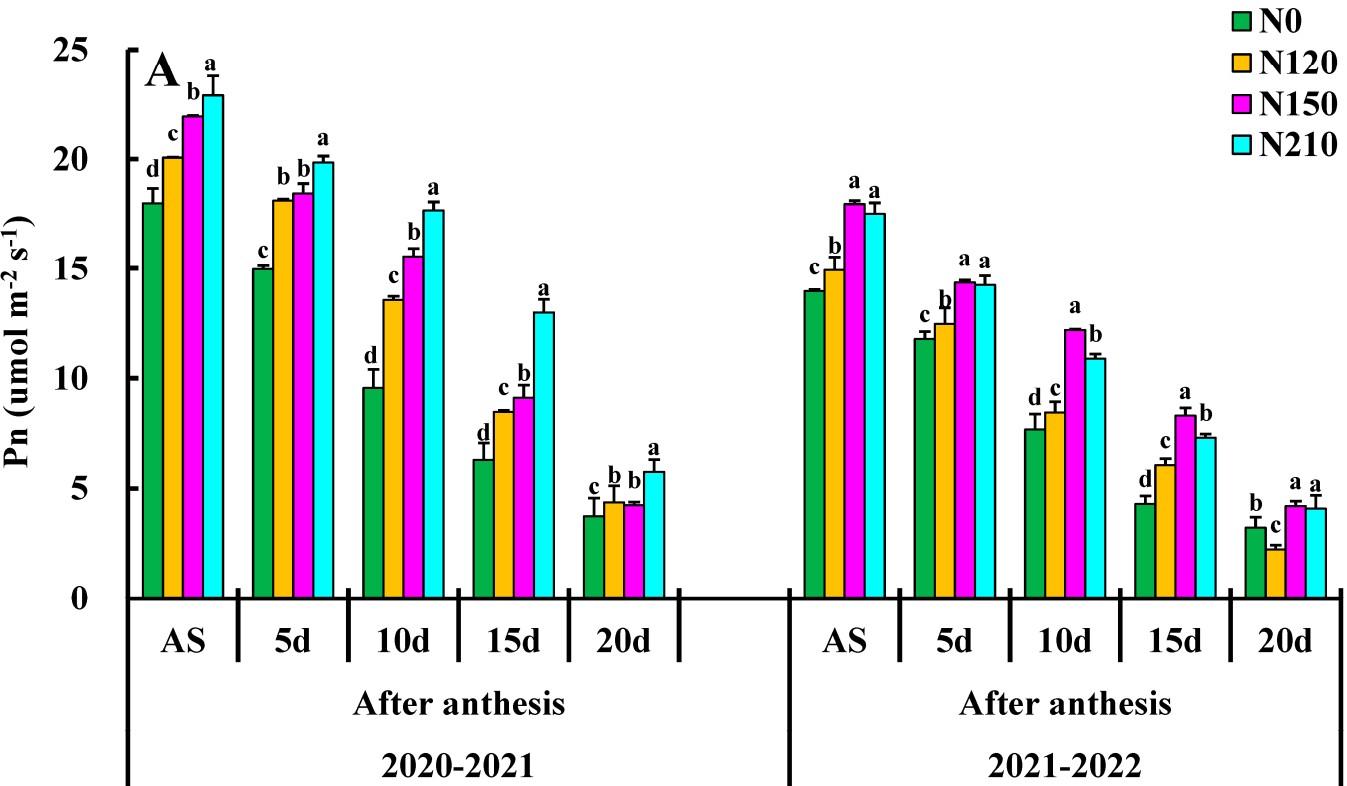

**Figure 3.** *Cont.*

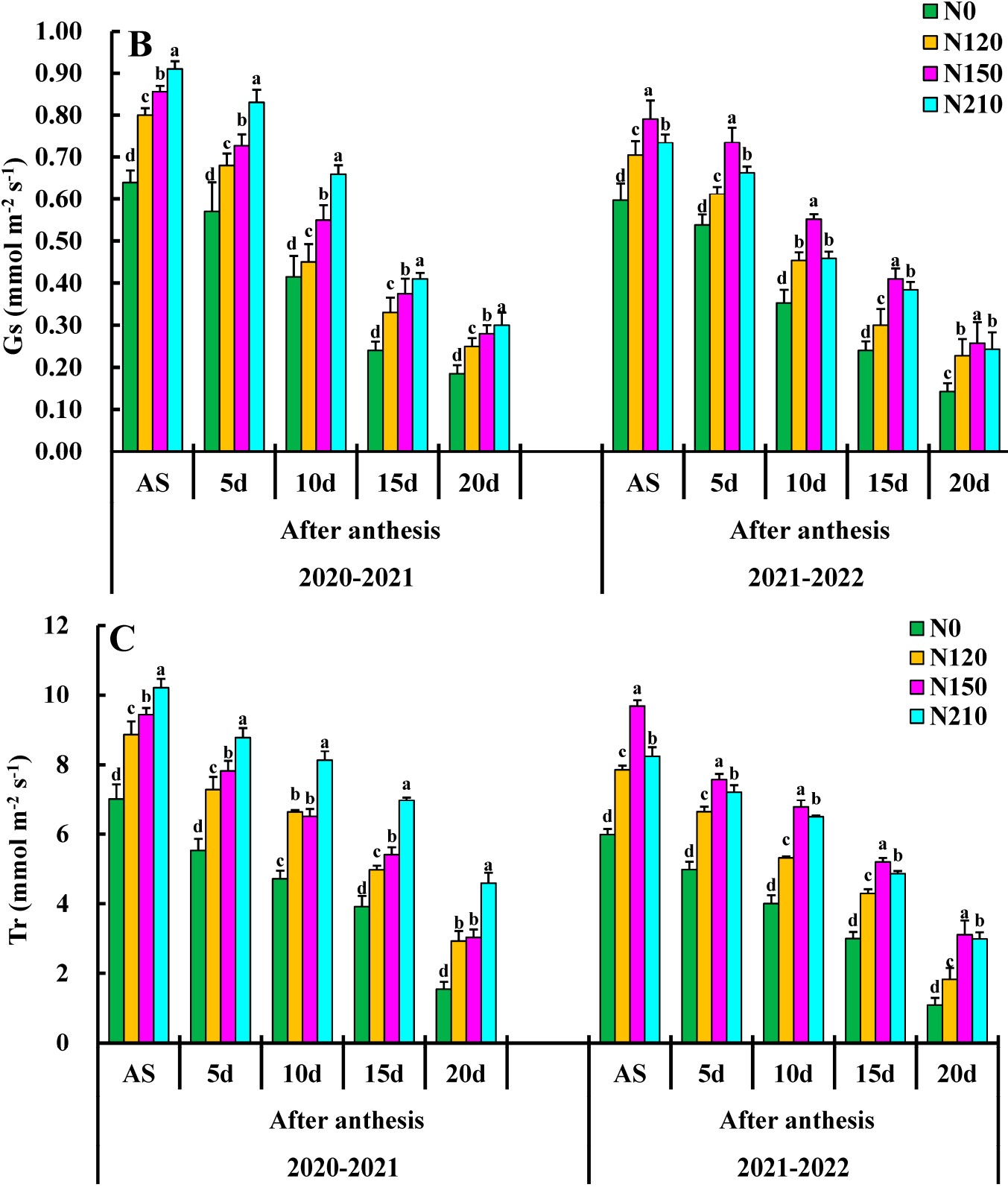

**Figure 3.** *Cont.*

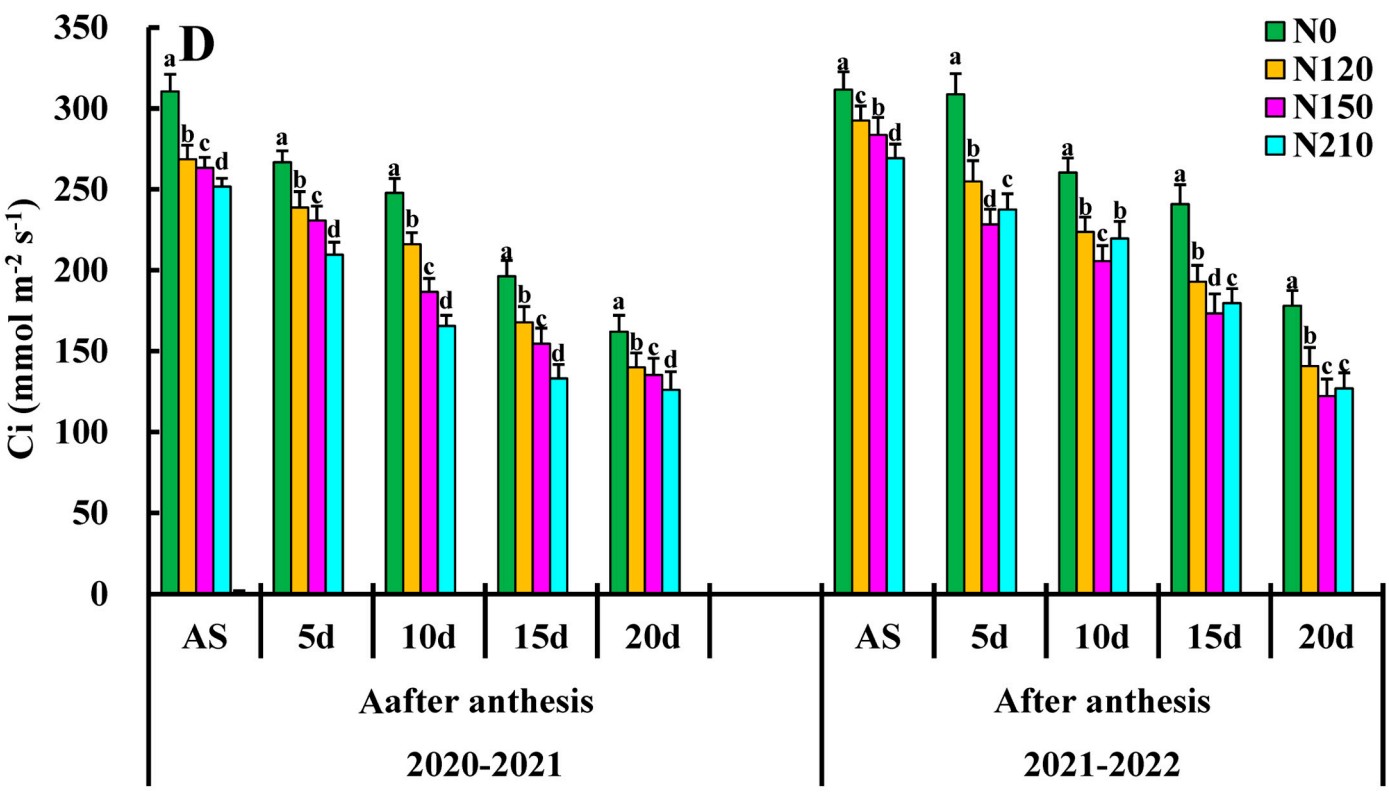

**Figure 3.** Effects of nitrogen application on net photosynthetic rate ($P_N$), stomatal conductance (gs), transpiration rate (Tr), and Ci: intercellular $CO_2$ concentration. Different lower cases indicate significant differences among treatments.

The effects of different nitrogen applications are shown in Figure 3C, with the transpiration rate (Tr) mmol ($H_2O$) m$^{-2}$ s$^{-1}$ of each treatment demonstrating an overall trend of Tr first increasing and then decreasing, respectively. Tr leaves were increased slowly from the days after the anthesis average value increased by 78%, reaching the maximum value at the flowering stage. The average growth from days after anthesis was 4.30 mmol m$^{-2}$ s$^{-1}$. The Tr value of N120 and N150 on the days after anthesis was greater than that of N0 and N210, and there was no significant difference between N0 and N120. The Tr value of 10–20 d treatment was significantly higher than that of nitrogen treatment on the days after anthesis and reached the maximum value at N150 and N210, except for the nitrogen application. N0 had no significant effects on Tr mmol ($H_2O$) m$^{-2}$ s$^{-1}$, the nitrogen application. The interaction effects of nitrogen application rate 150 on Tr were the largest on the days after anthesis, so optimizing the combination of nitrogen application rate could enhance the leaf function of wheat at the late stage.

The effects of different nitrogen applications and the influence of the Ci winter wheat flag leaves are shown in Figure 3D. The maximum Ci value of days was 505.28 Ci (μmol ($CO_2$) mol$^{-1}$) with the increased nitrogen application rate, and the Ci value decreased significantly. The Ci value of 10–20 d, N120 treatment at the days after anthesis, and N0 treatment might be due to the high nitrogen level. Environmental stress on the flag leaf causes the stomata to close on the wheat flag leaf, increasing Ci. The N0 at days after the anthesis of growth had significant effects on the Ci winter wheat flag leaf, and the interaction between these two interactions was extremely significant days after the anthesis.

3.2.1. Regulation of Nitrogen Application on Aboveground Nitrogen Allocation

The different nitrogen fertilizers were provided in these years 2019–2020, different treatments compares to others nitrogen application and flowering leaves, as shown in Table 2. At the maturity stage, compared with the N0 treatment, increased N application

increased N in stem sheath, leaf, and glume in 2020–2021, and increased N application increased the leaf N allocation index and sheath N allocation index, except that N150 decreased the leaf N allocation index. With the increase in N fertilizer, the N allocation index of glume decreased slightly at first and then increased slightly, but N fertilizer had no significant effect on the N allocation index of grain. The N120 treatment was significantly higher than that of other treatments. With the increase in nitrogen fertilizer, the leaf N allocation index decreased slightly at first and then increased significantly. The nitrogen allocation index of glume increased with the increase in N fertilizer in 2021–2022, and the increase in N120 treatment reached a significant level. In grain, the increase in N fertilizer and the decrease of grain N allocation index reached a significant level under N210 treatment. In general, increased N application increased the N allocation index in stem sheath and leaf at the maturity stage, but decreased the N allocation index in grain at the maturity stage.

**Table 2.** Effects of N application on aboveground N partitioning at anthesis and physiological maturity for the three experimental years.

| Year | N Application (kg ha$^{-1}$) | Anthesis (%) | | | Harvest (%) | | | |
|---|---|---|---|---|---|---|---|---|
| | | Stems + Sheath | Leaves | Ears | Stems + Sheath | Leaves | Chaff | Grains |
| 2019–2020 | N0 | 36.08 [b] | 44.69 [a] | 19.23 [b] | 7.29 [c] | 6.82 [b] | 8.93 [a] | 76.96 [a] |
| | N120 | 42.82 [a] | 35.64 [b] | 21.54 [ab] | 9.69 [b] | 7.04 [b] | 7.80 [b] | 75.47 [a] |
| | N150 | 37.57 [b] | 39.02 [b] | 23.41 [a] | 11.07 [a] | 9.90 [a] | 7.61 [b] | 71.41 [a] |
| | N210 | 40.91 [ab] | 36.62 [b] | 22.46 [a] | 12.27 [a] | 9.35 [a] | 7.33 [b] | 71.05 [a] |
| 2020–2021 | N0 | 21.71 [b] | 54.99 [a] | 23.30 [a] | 12.00 [b] | 8.35 [b] | 14.08 [ab] | 65.57 [a] |
| | N120 | 23.97 [b] | 52.86 [ab] | 23.18 [a] | 12.25 [b] | 8.43 [b] | 13.26 [b] | 66.07 [a] |
| | N150 | 28.34 [a] | 46.84 [bc] | 24.82 [a] | 12.98 [b] | 7.08 [c] | 15.10 [a] | 64.84 [a] |
| | N210 | 30.99 [a] | 44.04 [c] | 24.97 [a] | 14.91 [a] | 9.93 [a] | 15.09 [a] | 60.07 [a] |
| 2021–2022 | N0 | 40.15 [b] | 33.27 [b] | 26.58 [ab] | 21.52 [ab] | 7.28 [b] | 9.35 [b] | 61.85 [a] |
| | N120 | 46.57 [a] | 25.90 [c] | 27.54 [a] | 23.14 [a] | 6.11 [b] | 10.84 [a] | 59.91 [ab] |
| | N150 | 39.10 [b] | 35.55 [ab] | 25.35 [ab] | 19.14 [b] | 13.49 [a] | 10.35 [ab] | 57.03 [ab] |
| | N210 | 36.70 [b] | 39.21 [a] | 24.08 [b] | 21.41 [ab] | 13.21 [a] | 10.45 [ab] | 54.94 [b] |
| Probability level of ANOVA | | | | | | | | |
| Y | | ** | ** | ns | *** | *** | *** | ** |
| N | | *** | *** | ** | *** | *** | ns | * |
| Y × N | | ** | *** | * | *** | ** | ** | ns |

Note: different lowercase letters represent significant differences between treatments for the same year ($p \leq 0.05$); *, **, *** indicated significant differences at $p = 0.05, 0.01, 0.001$ levels, respectively; ns: not significant.

3.2.2. Regulation of Pre-Flowering Nitrogen Transfer Volume and Transfer Efficiency by Nitrogen Fertilizer

The nitrogen transfer volume of stem sheath, leaf, and glume increased with the increase in nitrogen application rate (2019–2020) 0 (N0), 120 (N120), 210 (N210) kg ha$^{-1}$; treatments increased the whole above ground N transfer volume by 28.7%, 55.7%, and 96.9%, respectively. N120, N150, and N210 kg ha$^{-1}$ treatments (2020–2021) increased the nitrogen transshipment by 11.8%, 57.0%, and 70.7%, respectively (Table 3). N120 and N210 treatments increased N transfer volume by 24.7%, 99.0%, and 135.5%, respectively. On average, compared with N0, N120 and N210 increased the nitrogen transfer volume of stem sheath by 46.6%, 93.8%, and 124.8%, and leaf nitrogen transfer volume by 6%, respectively, as well as 31.5% and 52.9%, increased stems + sheath nitrogen transshipment by 40.4%, 125.8%, and 159.6%, respectively. N120, N150, and N210 treatments increased the aboveground nitrogen transfer by 14.8%, 62.2%, and 88.3%, respectively. The effects of nitrogen fertilizer on nitrogen transport efficiency depended on interannual rainfall. N0 treatment, the nitrogen transport efficiency of chaff and leaf decreased with the increase in nitrogen fertilizer in wet years (2021–2022), while the nitrogen transport efficiency of leaves

increased with the increase in nitrogen fertilizer. Treatment N120, N150, and N210 kg ha$^{-1}$ reduced aboveground nitrogen transport efficiency by 5.5%, 8.3%, and 4.5%, respectively, in wet years ($p > 0.05$). The nitrogen transport efficiency of stem sheath and the nitrogen transport efficiency of leaves increased first and then decreased. On average, treatments N150 and N210 increased nitrogen transport efficiency in stem sheath by 18.8% and 13.6% ($p < 0.05$). N remobilization efficiency increased by 25.8 and 31.1% ($p < 0.05$), and decreased leaf nitrogen transport efficiency by 10.7%, respectively. However, nitrogen fertilizer had no significant effect on the overall nitrogen transport efficiency ($p > 0.05$).

**Table 3.** Effects of N application on pre-anthesis stored N remobilization and remobilization efficiency for the three experimental years.

| Year | N Application (kg ha$^{-1}$) | N Remobilization NR (g m$^{-2}$) | | | | N Remobilization Efficiency NRE (%) | | | |
|---|---|---|---|---|---|---|---|---|---|
| | | Stems + Sheath | Leaves | Chaff | Sum | Stems + Sheath | Leaves | Chaff | Whole Aboveground |
| 2019–2020 | N0 | 3.24 [c] | 3.70 [c] | 0.89 [d] | 7.83 [d] | 72.88 [a] | 79.52 [a] | 37.66 [c] | 69.07 [a] |
| | N120 | 4.19 [b] | 4.38 [bc] | 1.51 [c] | 10.08 [c] | 67.95 [a] | 72.05 [ab] | 48.71 [b] | 65.27 [b] |
| | N150 | 4.50 [b] | 5.06 [b] | 2.63 [b] | 12.19 [b] | 62.23 [b] | 67.45 [b] | 58.30 [a] | 63.34 [b] |
| | N210 | 6.19 [a] | 5.99 [a] | 3.24 [a] | 15.42 [a] | 64.76 [ab] | 69.99 [b] | 61.63 [a] | 65.97 [ab] |
| 2020–2021 | N0 | 0.44 [d] | 3.61 [b] | 0.34 [d] | 4.40 [c] | 24.64 [c] | 79.31 [ab] | 17.65 [c] | 53.07 [a] |
| | N120 | 0.69 [c] | 3.74 [b] | 0.50 [c] | 4.92 [c] | 32.00 [b] | 78.79 [ab] | 23.90 [b] | 54.86 [a] |
| | N150 | 1.51 [b] | 4.58 [a] | 0.76 [b] | 6.84 [b] | 43.50 [a] | 81.35 [a] | 24.95 [b] | 56.63 [a] |
| | N210 | 1.91 [a] | 4.65 [a] | 1.02 [a] | 7.58 [a] | 43.48 [a] | 73.50 [b] | 28.95 [a] | 53.07 [a] |
| 2021–2022 | N0 | 1.15 [c] | 2.27 [c] | 1.43 [c] | 5.11 [b] | 28.13 [b] | 70.67 [a] | 52.87 [a] | 48.86 [a] |
| | N120 | 2.20 [b] | 2.40 [c] | 1.74 [b] | 6.34 [b] | 38.25 [ab] | 70.66 [a] | 51.05 [a] | 50.17 [a] |
| | N150 | 3.01 [a] | 3.93 [b] | 2.64 [a] | 9.91 [a] | 43.52 [a] | 56.21 [b] | 52.90 [a] | 50.41 [a] |
| | N210 | 3.34 [a] | 5.30 [a] | 2.68 [a] | 11.73 [a] | 34.44 [b] | 62.15 [b] | 51.24 [a] | 49.35 [a] |
| Probability level of ANOVA | | | | | | | | | |
| Y | | *** | *** | *** | *** | *** | *** | *** | *** |
| N | | *** | *** | *** | *** | ** | ** | *** | ns |
| Y × N | | ** | *** | ** | ** | ** | * | *** | ns |

Note: different lowercase letters represent significant differences between treatments for the same year ($p \leq 0.05$); *, **, *** indicated significant differences at $p$ = 0.05, 0.01, 0.001 levels, respectively; ns: not significant.

### 3.2.3. Regulation of Post-Flowering Dry Matter and Nitrogen Accumulation by Nitrogen Fertilizer

The effects of nitrogen fertilizer on biomass accumulation, post-flowering dry matter (DM) accumulation and post-flowering nitrogen uptake were dependent on rainfall years (Table 4). In general, biomass accumulation at flowering and maturity and dry matter accumulation after flowering increased with increasing nitrogen fertilizer. In 2019–2020, compared with N0 treatments, N120, N120, and N210 significantly increased aboveground biomass and post-flowering dry matter accumulation at the anthesis stage, while biomass accumulation at the maturity stage only reached a significant level under N150 and N210 treatments. Postanthesis N uptake was only significant under N120 treatment. When the N application rate was higher than N120, the postanthesis N accumulation decreased slightly with the increase in the N application rate. In 2020–2021, when the nitrogen application range was 150 kg ha$^{-1}$, the biomass accumulation at the flowering stage and maturity stage increased significantly with the increase in nitrogen fertilizer; when the nitrogen application rate was greater than N150, the biomass accumulation at the flowering stage and maturity stage did not increase significantly, while in 2021–2022, nitrogen fertilizer had no significant effect on postanthesis dry matter accumulation, but the postanthesis nitrogen accumulation increased with nitrogen fertilizer.

**Table 4.** Effects of N application on biomass accumulation at flowering and at physiological maturity, post-anthesis DM accumulation (PADMA), and post-anthesis N uptake (PANU) for the three experimental years.

| Year | N Application (kg ha$^{-1}$) | Biomass (Mg ha$^{-1}$) | | PADMA | PANU |
|---|---|---|---|---|---|
| | | **Anthesis** | **Maturity** | **Mg ha$^{-1}$** | **kg ha$^{-1}$** |
| 2019–2020 | N0 | 6.53 [d] | 10.65 [c] | 4.12 [c] | 42.20 [b] |
| | N120 | 8.32 [c] | 12.99 [c] | 4.67 [b] | 59.90 [a] |
| | N150 | 9.90 [b] | 14.91 [b] | 4.96 [b] | 54.30 [ab] |
| | N210 | 11.82 [a] | 17.37 [a] | 5.55 [a] | 41.00 [b] |
| 2020–2021 | N0 | 5.03 [c] | 7.12 [c] | 2.09 [a] | 30.10 [a] |
| | N120 | 6.19 [b] | 8.46 [b] | 2.27 [a] | 29.60 [a] |
| | N150 | 7.56 [a] | 9.61 [a] | 2.05 [a] | 28.50 [a] |
| | N210 | 7.7 [a] | 9.74 [a] | 2.04 [a] | 24.80 [b] |
| 2021–2022 | N0 | 8.27 [c] | 11.59 [b] | 3.32 [b] | 34.70 [a] |
| | N120 | 9.01 [b] | 12.45 [b] | 3.44 [b] | 30.10 [b] |
| | N150 | 10.62 [a] | 15.34 [a] | 4.72 [a] | 30.30 [b] |
| | N210 | 11.35 [a] | 15.62 [a] | 4.27 [a] | 27.00 [b] |
| Probability level of ANOVA | | | | | |
| Y | | *** | *** | *** | *** |
| N | | ** | ** | ** | ns |
| Y × N | | * | * | * | ns |

Note: different lowercase letters represent significant differences between treatments for the same year ($p \leq 0.05$); *, **, *** indicated significant differences at $p$ = 0.05, 0.01, 0.001 levels, respectively; ns: not significant.

### 3.2.4. Effects of Nitrogen Application on SPAD Leaf of Winter Wheat

The SPAD values in leaves of winter wheat and the SPAD value of the leaf of wheat in each treatment increased rapidly in a small range from the jointing stage to the booting stage, with an average growth rate of 13.2%, as shown in Table 5. The SPAD value of wheat leaves in each treatment from the booting stage to the flowering stage increased slowly with an average growth rate of only 2.14%. The average values were 2.91% higher than those of 10–15 days. The SPAD value of the N150 and N210 treatments was the highest. The average SPAD value of N150 was 48.81, which was 1.04 times that of N0 and 1.02 times N120. The amount of SPAD in leaves under different conditions varied according to the amount of nitrogen consumption on different days. N0 had no significant difference with N150 except that there was a significant difference between N150 and N210 at the booting stage.

### 3.2.5. Effects of Nitrogen Application Rate on Grain Protein

Nitrogen application on glutenin content, gluten to alcohol ratio, protein content, and yield. Nevertheless, the interaction of nitrogen application amount year and nitrogen application amount had extremely significant alcohol solubility, gluten content, glutenin to alcohol ratio, protein content and yield (Table 6). The contents of clear, spherical, alcohol-soluble, and glutenin in wheat grains were not significantly different, but the protein yield was significantly increased by 13.4–16.3%. With the increase in nitrogen application rate of 210 kg ha$^{-1}$, the contents of clear, pellet, alcohol solubility, gluten, gluten to alcohol ratio and protein showed an increasing trend. With the increase in nitrogen application rate (150 kg ha$^{-1}$), protein yield showed an increasing trend.

**Table 5.** Effects of nitrogen application on SPAD value of flag leaf of winter wheat.

| Year | N Application (kg ha$^{-1}$) | Growth Stage | | | |
|---|---|---|---|---|---|
| | | 0 Day | 5 Day | 10 Day | 15 Day |
| 2019–2020 | N0 | 46.38 [a] | 52.79 [c] | 53.1 [a] | 46.37 [a] |
| | N120 | 46.19 [a] | 53.28 [b] | 54.72 [a] | 47.97 [a] |
| | N150 | 47.47 [a] | 54.12 [a] | 55.75 [a] | 49.47 [a] |
| | N210 | 47.14 [a] | 53.17 [bc] | 55.62 [a] | 47.77 [a] |
| | Mean | 46.79 | 53.34 | 54.80 | 47.90 |
| 2020–2021 | N0 | 45.64 [b] | 51.40 [b] | 51.77 [a] | 45.67 [a] |
| | N120 | 45.75 [ab] | 52.34 [a] | 53.01 [a] | 46.07 [a] |
| | N150 | 46.25 [a] | 52.73 [a] | 53.19 [a] | 47.27 [a] |
| | N210 | 46.29 [a] | 52.79 [a] | 53.34 [a] | 47.57 [a] |
| | Mean | 45.98 | 52.32 | 52.83 | 46.65 |
| 2021–2022 | N0 | 43.88 [b] | 49.53 [c] | 50.97 [b] | 43.65 [a] |
| | N120 | 45.86 [a] | 49.84 [c] | 51.37 [b] | 45.64 [a] |
| | N150 | 45.3 [a] | 51.53 [a] | 52.77 [a] | 45.67 [a] |
| | N210 | 45.07 [a] | 50.54 [b] | 51.77 [ab] | 45.07 [a] |
| | Mean | 45.03 | 50.35 | 51.72 | 45.01 |
| Probability level of ANOVA | | | | | |
| | Y | * | ** | * | * |
| | N | ns | ** | * | ns |
| | Y × N | ns | * | ns | ns |

Note: within a column for each N rate, means followed by different lower-case letters are significantly different according to Tukey's HSD test (0.05). Within a column, upper-case letters indicate comparisons among * and **, significant at 0.01 and 0.05 probability levels, respectively; ns, indicates not significant at 0.05 probability level.

**Table 6.** Effects of nitrogen application on grain protein and component contents at maturity of wheat.

| Year | N Application (kg ha$^{-1}$) | Albumin (%) | Gliadin (%) | Glutenin (%) | Glu/Gli | Protein (%) |
|---|---|---|---|---|---|---|
| 2019–2020 | N0 | 1.60 [b] | 3.95 [b] | 3.85 [b] | 0.97 [b] | 12.08 [b] |
| | N120 | 2.01 [a] | 4.35 [a] | 4.28 [a] | 0.98 [b] | 13.13 [a] |
| | N150 | 2.11 [a] | 4.36 [a] | 4.32 [a] | 0.99 [ab] | 13.51 [a] |
| | N210 | 2.20 [a] | 4.34 [a] | 4.37 [a] | 1.01 [a] | 13.63 [a] |
| | Mean | 1.98 [A] | 4.25 [A] | 4.21 [A] | 0.99 [A] | 13.09 [A] |
| 2020–2021 | N0 | 1.52 [b] | 3.78 [b] | 3.63 [b] | 0.96 [b] | 11.69 [b] |
| | N120 | 1.69 [a] | 4.16 [a] | 4.18 [a] | 1.00 [ab] | 13.13 [a] |
| | N150 | 1.75 [a] | 4.15 [a] | 4.25 [a] | 1.02 [ab] | 13.55 [a] |
| | N210 | 1.76 [a] | 4.15 [a] | 4.22 [a] | 1.02 [a] | 13.62 [a] |
| | Mean | 1.68 [A] | 4.06 [A] | 4.07 [A] | 1.00 [A] | 13.00 [A] |
| 2021–2022 | N0 | 1.53 [c] | 3.78 [c] | 3.68 [c] | 0.97 [b] | 11.63 [c] |
| | N120 | 1.66 [b] | 4.16 [b] | 4.05 [b] | 0.97 [b] | 13.05 [b] |
| | N150 | 1.66 [b] | 4.19 [b] | 4.06 [b] | 0.97 [b] | 13.25 [b] |
| | N210 | 1.88 [a] | 4.35 [a] | 4.39 [a] | 1.01 [a] | 14.15 [a] |
| | Mean | 1.68 [A] | 4.12 [A] | 4.05 [A] | 0.98 [A] | 13.02 [A] |
| Probability level of ANOVA | | | | | | |
| | Y | ** | ** | ** | * | ** |
| | N | ns | ns | ns | ns | ns |
| | Y × N | ** | ** | ** | ** | ** |

Note: within a column for each N rate means followed by different lower-case letters are significantly different according to Tukey's HSD test (0.05). Within a column, upper-case letters indicate comparisons among * and **, significant at 0.01 and 0.05 probability levels, respectively; ns, indicates not significant at 0.05 probability level.

### 3.2.6. Effects of Nitrogen Application on Grain Yield and Yield Components

The nitrogen application amount had extremely significant effects on panicle number, ear number per spike, and yield, and the interaction of nitrogen application amount had extremely significant effects on spike number, grain number per ear, 1000-grain weight, and yield (Table 7). The increase in nitrogen application rate (150 kg ha$^{-1}$), panicle number, and yield grain number per spike showed an increasing trend, while 1000-grain weight showed a decreasing trend. With the increase in nitrogen application rate (150, 210 kg ha$^{-1}$), the spike number increased while the 1000-grain weight and grain number per spike decreased. The yield of N210 showed an increasing trend, while that of N210 kg ha$^{-1}$ showed a decreasing trend. At the same time, there was no significant difference in yield between 120 kg ha$^{-1}$ and 150 kg ha$^{-1}$.

**Table 7.** Effects of nitrogen application on grain yield and yield components in growth period of wheat.

| Year | N Application (kg ha$^{-1}$) | Ear Number (10$^4$ ha$^{-1}$) | Grain Number per Ear | 1000-Grain Weight (g) | Yield (kg ha$^{-1}$) |
|---|---|---|---|---|---|
| | N0 | 358.3 [d] | 28.7 [c] | 42.1 [a] | 3868.9 [c] |
| | N120 | 500.5 [c] | 30.4 [a] | 39.9 [b] | 4450.8 [b] |
| 2019–2020 | N150 | 540.5 [b] | 32.1 [a] | 39.8 [b] | 6095.4 [a] |
| | N210 | 600.3 [a] | 31.3 [b] | 38.1 [c] | 5225.6 [a] |
| | Mean | 499.9 [A] | 31.1 [B] | 40.0 [A] | 7185.2 [A] |
| | N0 | 434.0 [d] | 23.1 [c] | 43.1 [a] | 3671.8 [c] |
| | N120 | 577.5 [c] | 25.0 [a] | 41.6 [b] | 4000.8 [b] |
| 2020–2021 | N150 | 613.0 [b] | 27.8 [a] | 41.1 [b] | 6593.2 [a] |
| | N210 | 680.1 [a] | 26.5 [b] | 40.8 [c] | 5778.5 [a] |
| | Mean | 576.2 [A] | 30.6 [B] | 41.6 [A] | 5511.1 [A] |
| | N0 | 290.3 [d] | 29.9 [c] | 42.5 [a] | 3450.1 [c] |
| | N120 | 438.0 [c] | 30.2 [a] | 40.2 [b] | 4857.0 [b] |
| 2021–2022 | N150 | 462.3 [b] | 32.9 [a] | 40.1 [b] | 6340.7 [a] |
| | N210 | 481.3 [a] | 31.3 [b] | 37.1 [c] | 5721.4 [b] |
| | Mean | 417.9 [B] | 33.3 [A] | 40.0 [A] | 6317.3 [B] |
| Probability level of ANOVA | | | | | |
| Y | | ** | ** | ** | ** |
| N | | ** | ** | ** | ** |
| Y × N | | ns | ns | ns | ns |

Note: within a column for each N rate means followed by different lower-case letters are significantly different according to Tukey's HSD test (0.05). Within a column, upper-case letters indicate comparisons among * and **, significant at 0.01 and 0.05 probability levels, respectively; ns, indicates not significant at 0.05 probability level.

### 3.2.7. Effects of Nitrogen Application on Starch Contents

The year and nitrogen application amount had significant or extremely significant effects on amylose, amylopectin, and total starch content; the straight/branch and nitrogen application amount had extremely significant effects on starch yield; and with the interaction of nitrogen application amount, nitrogen application N0 had extremely significant effects on amylose, amylopectin, total starch content, and straight/branch (Table 8). The contents of amylose, amylopectin, total starch, and straight/branch of wheat grains in N120 had no significant difference but significantly increased the starch yield by 14.7–16.4%. The yield of amylose, amylopectin, total starch, and direct/branch ratio decreased with the increase in nitrogen application rate N210, while the yield of starch in N150 increased first and then decreased, while the yield of starch in N120 showed an increasing trend.

**Table 8.** Effects of nitrogen application on starch contents in growth period of wheat.

| Year | N Application (kg ha$^{-1}$) | Am (%) | Ap (%) | Starch (%) | Am/Ap | Sy (kg ha$^{-1}$) |
|---|---|---|---|---|---|---|
| 2019–2020 | N0 | 17.5 [a] | 57.7 [a] | 75.2 [a] | 0.30 [a] | 3875.8 [c] |
| | N120 | 14.1 [b] | 57.9 [a] | 72.0 [b] | 0.24 [b] | 5737.6 [a] |
| | N150 | 13.4 [bc] | 51.8 [b] | 65.1 [c] | 0.26 [b] | 5496.4 [a] |
| | N210 | 13.1 [c] | 51.4 [b] | 64.5 [c] | 0.25 [b] | 5044.2 [b] |
| | Mean | 14.5 [A] | 54.7 [A] | 69.2 [A] | 0.26 [A] | 5038.5 [B] |
| 2020–2021 | N0 | 17.5 [a] | 58.0 [a] | 75.4 [a] | 0.30 [a] | 4278.2 [b] |
| | N120 | 14.2 [b] | 57.7 [a] | 71.8 [b] | 0.25 [b] | 6465.3 [a] |
| | N150 | 13.4 [bc] | 52.4 [b] | 65.7 [c] | 0.26 [b] | 6303.7 [a] |
| | N210 | 13.3 [c] | 52.3 [b] | 65.6 [c] | 0.25 [b] | 6416.6 [a] |
| | Mean | 14.6 [A] | 55.1 [A] | 69.6 [A] | 0.26 [A] | 5866.0 [A] |
| 2021–2022 | N0 | 15.5 [a] | 54.3 [a] | 69.8 [a] | 0.29 [a] | 3122.2 [c] |
| | N120 | 13.6 [b] | 54.1 [a] | 67.7 [b] | 0.25 [b] | 4640.8 [a] |
| | N150 | 13.5 [b] | 50.3 [b] | 63.7 [c] | 0.27 [b] | 4676.0 [a] |
| | N210 | 12.1 [c] | 50.5 [b] | 62.6 [c] | 0.24 [b] | 4209.6 [b] |
| | Mean | 14.0 [A] | 52.3 [A] | 65.9 [A] | 0.26 [A] | 4162.1 [B] |
| Probability level of ANOVA | | | | | | |
| Y | | ** | ** | ** | * | * |
| N | | ** | ** | ** | ** | ** |
| Y × N | | ns | ns | ns | ns | ns |

Note: Am: amylose content; Ap: amylopectin content; Am/Ap: amylose content/amylopectin content; Sy: starch yield. Within a column for each nitrogen rate means followed by different lower-case letters are significantly different according to Tukey's HSD test (0.05). Within a column, upper-case letters indicate comparisons among * and **, significant at 0.01 and 0.05 probability levels, respectively; ns, indicates not significant at 0.05 probability level.

### 3.2.8. Correlation Analysis of Yield with Photosynthesis in Flag Leaves of Wheat

The yield and post-anthesis photosynthesis at maturity were significantly correlated with net photosynthetic, superoxide dismutase activities (SOD), peroxidase isozyme (POD), malondialdehyde concentration (MDA), soluble protein concentration (SPC), catalase (CAT) of wheat flag leaves at 10–20 days after anthesis (Table 9), and with the photosynthesis of wheat flag leaves at 10–20 days after anthesis. In conclusion, photosynthesis and fluorescence in the middle and late growth stages were beneficial to the accumulation of dry matter and nitrogen and promoted the increase in yield.

**Table 9.** Correlation analysis of photosynthetic characteristics with yield and dry matter.

| Index | | 0 Day | 5 Day | 10 Day | 15 Day | 20 Day |
|---|---|---|---|---|---|---|
| Yield | | 0.4212 | 0.4655 | 0.5612 * | 0.8212 ** | 0.9125 ** |
| SOD | Pn | 0.4259 | 0.4545 | 0.6419 * | 0.8512 ** | 0.9645 ** |
| POD | | 0.4825 | 0.4745 | 0.5819 * | 0.8716 ** | 0.9569 ** |
| Yield | | 0.4555 | 0.4268 | 0.6519 * | 0.8215 ** | 0.8155 ** |
| MDA | Gs | 0.4561 | 0.4556 | 0.6412 * | 0.9112 ** | 0.9645 ** |
| SPC | | 0.4666 | 0.3755 | 0.6535 * | 0.8612 ** | 0.8555 ** |
| Yield | | 0.4557 | 0.3612 | 0.6602 * | 0.7126 ** | 0.7115 ** |
| CAT | Tr | 0.4565 | 0.3566 | 0.6802 * | 0.9125 ** | 0.8655 ** |
| POD | | 0.4569 | 0.4765 | 0.6836 * | 0.8226 ** | 0.8555 ** |
| Yield | | 0.4066 | 0.4655 | 0.5966 * | 0.7155 ** | 0.8147 ** |
| DM | Ci | 0.4666 | 0.4555 | 0.6555 * | 0.8154 ** | 0.8555 ** |
| MDA | | 0.4554 | 0.4612 | 0.5966 * | 0.8612 ** | 0.8109 ** |

Note: * and ** denote significant correlations at 5% and 1% probability levels, respectively. superoxide dismutase (SOD); peroxidase isozyme (POD); malondialdehyde concentration (MDA); soluble protein concentration (SPC); catalase (CAT); dry matter (DM), in wheat flag leaves. Net photosynthetic rate (Pn); stomatal conductance (gs); transpiration rate (Tr); and Ci: intercellular $CO_2$ concentration.

## 4. Discussion

### 4.1. Effects of Different Nitrogen Applications on Photosynthesize Characteristics

Photosynthesis is usually limited under drought stress, and excess light energy tends to induce photoinhibition and even produces photooxidation, causing harm to the photosynthetic membrane system to occur [24]. Malondialdehyde (MDA) secondary product of membrane lipid peroxidation is usually used to indicate the degree of oxidative damage suffered by the membrane system under stress [25,26]. In response to oxidative damage in times of stress, plants tend to produce a range of oxygen chemical defense systems, including superoxide dismutase (SOD), catalase (CAT), and peroxidase (POD) [27]. Many studies have shown that the activity of these antioxidant enzymes is related to the ability of plants to withstand abiotic stresses, such as drought stress and salt stress [28,29]. In the experiment, the decrease in MDA content and the increase in SOD and POD activities promoted the formation of yield [26,30]. The increase in MDA content often indicates an increase in oxidation damage to the membrane system, which will affect the normal junction of membrane system structure and function [31,32]. Drought stress increased MDA content; however, N fertilizers replacing part of nitrogen fertilizer had lower MDA content than the control [33,34]. The leaf is an important component of the later growth stage, contributing about 30% to the photosynthesis of the population of wheat [35] and about 20% and 30% of dry matter in wheat grains comes from the photosynthesis of the parietal leaf. Therefore, the photosynthetic characteristics of winter wheat leaves can represent the photosynthetic characteristics of winter wheat, to a certain extent [36]. The maximum values of $P_N$ in the parietal leaves of winter wheat under the N rate were 18.46 $\mu mol\ m^{-2}\ s^{-1}$, 20.69 [37]. Nitrogen application contributed to the simultaneous improvement of carbon and nitrogen metabolism in wheat plants. The increase in nitrogen application rate ($P_N$) in the leaf of winter wheat increased first and then decreased with the increase in nitrogen application rate when the nitrogen application rate was N150 kg ha$^{-1}$, which was similar to that of the leaf as an important component of wheat at the later growth stage [38]. The main channel for the diffusion of $CO_2$ and water vapor inside and outside leaves plays an important role in regulating the transpiration and photosynthetic physiological process of leaves, and the growth stage [39]. A good understanding of the response of photosynthesis rate ($P_N$) and transpiration rate (Tr) to stomatal alteration during the diurnal variations is important to cumulative photosynthetic production and water loss of crops [40]. Stomatal conductance and intercellular $CO_2$ concentration of winter wheat in the late flowering stage were always higher than those in the early flowering period [41]. The $gs$, Tr, and $P_N$ of wheat have similar change rules, and there is a positive correlation between them [42]. The population structure was more reasonable, which played a significant role in the effective utilization of light energy and the improvement of grain yield. Flag leaf SPAD value and photosynthetic rate of wheat planted with N150 were higher, yield components were more reasonable, and yield was significantly higher than other varieties [43]. Nitrogen application of 150 kg ha$^{-1}$ could significantly enhance the photosynthetic efficiency of wheat, improve the problems of greenhouse diseases and insect pests at the later growth stage, and the photosynthetic rate improvement of wheat yield. Some interesting results were obtained in this study including nitrogen application of 150 kg ha$^{-1}$ could significantly improve the area of single stem leaf and light energy utilization efficiency of wheat. Additionally, it could increase wheat grain weight at the later growth stage compared to N90 kg ha$^{-1}$ [23]. The leaf area index of winter wheat increased under N150, especially in the middle- and late-growth stages. The differences in leaf area under the different nitrogen applications could be attributed to the differences in the wheat distribution. Wheat with N150 intercepts light more properly with a higher photosynthesis rate and leaf area index [44], the stored nitrogen of all dryland wheat varieties could not meet the requirements of nitrogen transport of crops, and photosynthetic nitrogen transport was carried out at the average photosynthetic nitrogen transport rate of 87.5%, which was significantly higher than that of N150 kg ha$^{-1}$ 69.5% [45]. The transpiration rate (Tr) and net photosynthetic rate ($P_N$) of N150 decreased significantly when the grain weight, yield, and grain starch content of wheat decreased significantly. The

premise of further reducing the amount of nitrogen application, at the jointing stage could improve grain quality, and increase nitrogen-use efficiency [46]. Nitrogen is an important component of plants and various dry gluten and approximately 75% of the nitrogen in plant leaves, most of which is used to construct photosynthetic apparatus [47]. It has been reported that the long-term response to $CO_2$ concentrations is mainly due to the limitation of nitrogen supply [48]. Many experiments have also detected a decrease in plant nitrogen concentration under $CO_2$ enrichment conditions [49]. The variation coefficient of chlorophyll fluorescence parameters under 7 d of water at the booting stage was higher than that under the compensation effect of nitrogen fertilizer, which was earlier than that under the control without N150. The late-growth functional leaves of wheat had stronger light capture ability and photochemical efficiency, which improved the photosynthetic performance of flag leaves [50]. The results showed that $P_N$, Tr, and intercellular $CO_2$ concentration ($C_i$) in the leaves of winter wheat had a similar trend to that of $g$s, and the values were the highest at the flowering stage. The SPAD value of burrow leaves under N150 treatment was higher, which might be because the population structure under N150 treatment was more conducive to the accumulation of photosynthesis of winter wheat, thus promoting the synthesis of chlorophyll and improving the SPAD value of flag leaves of winter wheat. The present study showing improvement of photosynthesis characteristics and the growth of N150 kg ha$^{-1}$ improved photosynthesis efficiency growth, which reached a significant level. The grain protein content of N150 could meet the standard of high-quality strong gluten, and the yield performance of N150 kg ha$^{-1}$ was significantly different from that of high yield.

### 4.2. Effects of Different Nitrogen Applications on Yield Formation

The nitrogen uptake ratio of different special types of wheat was different in each growth period, and the nitrogen uptake and nitrogen uptake ratio of wheat was the largest from jointing to flowering [12]. The retransfer of wheat nitrogen from vegetative organs to grains is restricted by environmental factors and photosynthetic differences. In the process of nitrogen application, different varieties will not only affect the effects of nitrogen application, but also cause the difference in nitrogen absorption, accumulation, and transport, as well as dry matter accumulation, distribution, and yield formation of crops. Therefore, only the combination of appropriate nitrogen regulation and rational planting measures to coordinate the contradiction between crop nutrient absorption and nitrogen-use efficiency can improve the nitrogen-use efficiency of wheat, and achieve high yield [51]. Some studies have suggested that nitrogen level has a significant impact on nitrogen uptake and the amount of nitrogen accumulated in the aboveground increases significantly with the increase in nitrogen level [52]. We divided the tested materials into high nitrogen efficiency, medium nitrogen efficiency, and low nitrogen efficiency based on nitrogen improving photosynthesis efficiency and nitrogen agronomic efficiency. Compared to medium nitrogen efficiency, it had higher yield, dry matter, and nitrogen accumulation [53]. The growth of plants above and below ground, relatively independent, and interdependent. The roots absorb water nutrients to meet the growth and development of the shoot, while the photosynthetic of the shoot is transported to the underground part through the stem, providing energy for root activities and meeting the material requirements of root activities [54]. The increase in nitrogen accumulation in the panicle would provide a good material basis for grain development, which promoted the assimilation ability of the source and increased the potential grain number, which was beneficial to yield formation [55]. Although water significantly inhibited post-anthesis dry matter and nitrogen accumulation, it stimulated plants to allocate more assimilates to grains, especially promoting the transport of pre-anthesis dry matter to grains [46]. The increase in nitrogen uptake after anthesis delayed the senescence of photosynthetic organs and improved photosynthetic capacity at the grain filling stage, thus increasing the accumulation of photosynthetic compounds after anthesis and ultimately improving grain yield [56]. Studies in wheat showed that nitrogen application combined with organic fertilizer increased nitrogen uptake and biomass accumulation,

thus increasing grain yield [57]. N treatment (N150 kg N ha$^{-1}$) and N fertilizer application 240 kg ha$^{-1}$ significantly increased dry matter and nitrogen accumulation after anthesis, but decreased biomass and nitrogen accumulation at anthesis, thus reducing the amount of nitrogen stored before anthesis to grain transport [58]. Optimize doses of nitrogen to increase the yield of the winter wheat crops. Thus, we used four nitrogen application concentrations set as N120 and N150, respectively, to analyze the photosynthetic characteristics and yield-related traits in winter wheat. Therefore, the SPAD with N150 kg ha$^{-1}$ would be a valuable management practice to improve wheat yield [59]. The partial replacement of nitrogen fertilizer by organic fertilizer promoted the distribution of nitrogen to panicle organs before another, thus increasing the pool size number of grains, and ultimately increasing the nitrogen requirement during grain filling, thus increasing the amount of post-anthesis nitrogen absorption [60]. It is necessary to further study the differences in the photosynthesis characteristics of N150 improvements related to nitrogen accumulation and transport, and the leaf expression characteristics among wheat with different nitrogen. Therefore, quantitative analysis of structural nitrogen, photosynthetic nitrogen, and storage nitrogen is of great significance for determining nitrogen accumulation and translocation in wheat plants. In this experimental work, the results showed that compared to other dryland wheat, N fertilizer N150 had higher structural nitrogen 53.42 kg ha$^{-1}$ on average, and N210 had higher structural nitrogen (N150 kg ha$^{-1}$), indicating that it had higher structural nitrogen than other N fertilizers.

## 5. Conclusions

In this experimental study, it is concluded that the malondialdehyde (MDA) of wheat flag leaves decreased with the increase in nitrogen application rate. The activities of superoxide dismutase (SOD) and peroxidase (POD) and soluble protein and catalase (CAT) in the kernel flag leaf of wheat after anthesis first increased and then decreased with the progress of grout. N fertilizer N150 had higher structural nitrogen, spike number and grain number per spike, as well as $P_N$, Tr, and intercellular $CO_2$ concentration ($C_i$) in the leaves, wherein yields were the highest compared to other N fertilizers.

**Author Contributions:** Conceptualization, H.N. and M.S.; methodology, M.S.; software, P.D.; validation, H.N., A.R. and Z.G.; formal analysis, Z.G.; investigation, M.S.; resources, H.N.; data curation, H.N.; writing—original draft preparation, H.N.; writing—review and editing, P.D.; visualization, A.R.; supervision, M.S.; project administration, M.S.; funding acquisition, H.N. All authors have read and agreed to the published version of the manuscript.

**Funding:** The authors are thankful to China Agriculture Research System (No. CARS-03-01-24), the National Natural Science Foundation of China (No. 32272216), the technology innovation team of Shanxi Province (No. 201605D131041), the Key Laboratory of Shanxi Province (No. 201705D111007), and the "1331" Engineering Key Laboratory of Shanxi Province for financial support of this study.

**Data Availability Statement:** Not applicable.

**Conflicts of Interest:** The authors declare no conflict of interest.

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
