# Peer review of "Effects of Nitrogen Fertilizer on Photosynthetic Characteristics and Yield"

_agronomy, doi:10.3390/agronomy13061550_

Round 1

Reviewer 1 Report (Previous Reviewer 1)

Dear Editor

MDPI Agronomy

I am sharing my observations about the manuscript: “Effects of Nitrogen Fertilizer on Photosynthetic Characteristics and Yield”. The authors did some important alterations, but objectives still aren’t clear. In my opinion, the author should review your statistical analysis, because they have four nitrogen rates (regression analysis). English continues very difficult to understand. Thus, my suggestion is to review English language and style.

Best regards.

23 May 2023.

The English language is hard to understand. Some paragraphs didn't make sense.

Author Response

agronomy-2437638

Effects of Nitrogen Fertilizer on Photosynthetic Characteristics and Yield

Dear Reviewer

Thank you very much for handling our manuscript. Thank you for your suggestion. Thank you for your decision and constructive comments on our manuscript. We had considered your suggestion carefully and made some changes. In the revised manuscript, we have attempted to address all the comments from the reviewers. All the changes that we have made in the revised manuscript. Below are our detailed point-to-point responses to the comments from the reviewers.

Thank you for your kind consideration.

Best regards,

Dr. Hafeez Noor

Key Laboratory of Functional Agriculture in the Loess Plateau, Ministry of Agriculture and Rural Affairs, Taigu,

College of Agronomy, Shanxi Agricultural University,

Taigu, Shanxi, 030801, P. R China

*********************************************************************

Review Report 1

I am sharing my observations about the manuscript: “Effects of Nitrogen Fertilizer on Photosynthetic Characteristics and Yield”. The authors did some important alterations, but objectives still aren’t clear. In my opinion, the author should review your statistical analysis, because they have four nitrogen rates (regression analysis). English continues very difficult to understand. Thus, my suggestion is to review English language and style.

Response: Thank you for your suggestion. The manuscript has benefited from these insightful suggestions. Thank you for your decision and constructive comments on our manuscript. We had considered your suggestion carefully and made some changes. In the revised manuscript, we have attempted to address all the comments from the reviewers. Below are our detailed point-to-point responses to the comments from the reviewers.

Reviewer 2 Report (Previous Reviewer 3)

Dear authors! you did a good job with the material. I ask you to pay attention to some shortcomings in the text. besides carefully compare the abstract and the conclusion. Thank you

Needs to be improved

Author Response

agronomy-2437638

Effects of Nitrogen Fertilizer on Photosynthetic Characteristics and Yield

Dear Reviewer

Thank you very much for handling our manuscript. Thank you for your suggestion. Thank you for your decision and constructive comments on our manuscript. We had considered your suggestion carefully and made some changes. In the revised manuscript, we have attempted to address all the comments from the reviewers. All the changes that we have made in the revised manuscript. Below are our detailed point-to-point responses to the comments from the reviewers.

Thank you for your kind consideration.

Best regards,

Dr. Hafeez Noor

Key Laboratory of Functional Agriculture in the Loess Plateau, Ministry of Agriculture and Rural Affairs, Taigu,

College of Agronomy, Shanxi Agricultural University,

Taigu, Shanxi, 030801, P. R China

*********************************************************************

Review Report-2

Dear authors! You did a good job with the material. I ask you to pay attention to some shortcomings in the text. Besides carefully compare the abstract and the conclusion. Thank you

Response: Thank you for your suggestion. The manuscript has benefited from these insightful suggestions. Thank you for your decision and constructive comments on our manuscript. We had considered your suggestion carefully and made some changes. In the revised manuscript, we have attempted to address all the comments from the reviewers. Below are our detailed point-to-point responses to the comments from the reviewers.

This manuscript is a resubmission of an earlier submission. The following is a list of the peer review reports and author responses from that submission.

Round 1

Reviewer 1 Report

Dear Editor

MDPI Agronomy

I am sharing my observations about the manuscript: “Nitrogen Fertilizer on Physiology Photosynthetic and chlorophyll fluorescence of Yield Food Nutrition Wheat (Triticum aestivum L.). The manuscript evaluated the effects of nitrogen in physiology and agronomic characteristics of the wheat. The results of the manuscript could help other studies on net photosynthetic rate (Pn), stomatal conductance (gs), transpiration rate (E), and Ci: intercellular CO2 concentration. However, the manuscript has lack of important information. The description from field experimental is uncompleted. The statistical analysis should be review, because although the authors have four nitrogen rates in your analysis they didn’t do regression analysis. Furthermore, the analysis of the interactions and isolated factors didn’t write in Material and Methods. The results have to show the results from statistical analysis, but sometimes the Table showed significance for interaction but in the manuscript the interactions isn’t written. The conclusions seem results. I don't feel qualified to judge about the English language and style, but there are some paragraphs hard to understand. Doubts and suggestions are in the manuscript.

 Best regards.

03 May, 2023.

Author Response

Agronomy-2404439

Effects of Nitrogen Fertilizer on Photosynthetic and chlorophyll fluorescence Yield of Wheat (Triticum aestivum L.)

Dear Reviewer

Thank you very much for handling our manuscript. Thank you for your suggestion. Thank you for your decision and constructive comments on our manuscript. We had considered your suggestion carefully and made some changes. In the revised manuscript, we have attempted to address all the comments from the reviewers. All the changes that we have made in the revised manuscript are highlighted with track Changes in the track file. Below are our detailed point-to-point responses to the comments from the reviewers.

Thank you for your kind consideration.

Best regards,

Dr. Hafeez Noor

Key Laboratory of Functional Agriculture in the Loess Plateau, Ministry of Agriculture and Rural Affairs, Taigu,

College of Agronomy, Shanxi Agricultural University,

Taigu, Shanxi, 030801, P. R China

*********************************************************************

Review Report 1

Dear Editor

I am sharing my observations about the manuscript: “Nitrogen Fertilizer on Physiology Photosynthetic and chlorophyll fluorescence of Yield Food Nutrition Wheat (Triticum aestivum L.). The manuscript evaluated the effects of nitrogen in physiology and agronomic characteristics of the wheat. The results of the manuscript could help other studies on net photosynthetic rate (Pn), stomatal conductance (gs), transpiration rate (E), and Ci: intercellular CO2 concentration. However, the manuscript has lack of important information. The description from field experimental is uncompleted. The statistical analysis should be review, because although the authors have four nitrogen rates in your analysis they didn’t do regression analysis. Furthermore, the analysis of the interactions and isolated factors didn’t write in Material and Methods. The results have to show the results from statistical analysis, but sometimes the Table showed significance for interaction but in the manuscript the interactions isn’t written. The conclusions seem results. I don't feel qualified to judge about the English language and style, but there are some paragraphs hard to understand. Doubts and suggestions are in the manuscript.

Response: Thank you for your suggestion. The manuscript has benefited from these insightful suggestions. Thank you for your decision and constructive comments on our manuscript. We had considered your suggestion carefully and made some changes. In the revised manuscript, we have attempted to address all the comments from the reviewers. Below are our detailed point-to-point responses to the comments from the reviewers.

Reviewer 2 Report

Dear authors, 

thank you for having submitted a nice manuscript on a topic of focal interest. I completely agree with the general outline of your experimental approach. However, I miss information that would allow transferring your results to other locations. In its current form, the manuscript is merely an interesting case study. However, the importance of your work can be easily improved by adding a few extra data, such as: 

  • Information on light intensities during the experiments. Average light intensity at noon, or the typical course of light intensity during a day would be sufficient. This would allow estimating how absorbed light energy is used (CO2 fixation, photorespiration, Asada cycle, etc.). The determination of the different fluorescence quenching types by PAM measurement are not sufficient to assess the positive effect of an electron sink that is mediated by nitrate reduction.
  • As a control you have used samples irrigated without added nitrogen fertilizer. But you did not provide information on the availability of ammonia or nitrate in the soil water phase of these controls.  
  • What type and what quality of N-fertilizer did you use?
  • What was the ion-exchange capacity of the soil? -> Did you measure the concentration of relevant ions in the soil water phase during the course of your experiments?

Finally: You have mentioned phosphate and potassium supply to the soil, but you do not provide any information on soil quality in general. Thus, your experiments can’t be reproduced elsewhere by colleagues. - Please improve.

lay out:

Fig 1   The figures are very small and therefore hard to read. Maybe they should be arranged in one column using page width.

Figures should be self-explaining. Therefore, explain shortages in use in the figure legend, please.

use of English language:

Obviously you have used a text interpreter. Nevertheless, the use of English language needs to be improved. Some words are not adequately translated. Therefore, it is hard to understand the meaning of some sentences (see the abstract, the conconclusion, for instrance). 

line 111 The term rate is not adequate, the meaning would be „change in nitrogen concentration per time interval“. You have applied an ammount of nitrogen fertilizer per hectar.

Fig 2   The term „nitrogen rate“ is not adequate  (see line 111)   -  In Table 1 you correctly are using the term „N application“. Please make sure to use this term in your manuscript throughout.

The conclusion chapter needs to be significantly improved. Sentences should be shorter, and all sentences need to contain a subject and a verb!  

content:

The complete discussion chapter is hard to read. Here not only the special use of English language is causing problems, but technical terms are not used correctly. SPAD, for instance, is a method to measure the concentration of photosynthetic active units per area. You are using this term to describe a chemical compound.

There is no clear line of arguments in the discussion subchapters, and some of the points you make are trivial.

The complete discussion chapter should be re-written in an understandable style.

Author Response

Agronomy-2404439

Effects of Nitrogen Fertilizer on Photosynthetic and chlorophyll fluorescence Yield of Wheat (Triticum aestivum L.)

Dear Reviewer

Thank you very much for handling our manuscript. Thank you for your suggestion. Thank you for your decision and constructive comments on our manuscript. We had considered your suggestion carefully and made some changes. In the revised manuscript, we have attempted to address all the comments from the reviewers. All the changes that we have made in the revised manuscript are highlighted with track Changes in the track file. Below are our detailed point-to-point responses to the comments from the reviewers.

Thank you for your kind consideration.

Best regards,

Dr. Hafeez Noor

Key Laboratory of Functional Agriculture in the Loess Plateau, Ministry of Agriculture and Rural Affairs, Taigu,

College of Agronomy, Shanxi Agricultural University,

Taigu, Shanxi, 030801, P. R China

*********************************************************************

Review Report-2

Dear authors

Thank you for having submitted a nice manuscript on a topic of focal interest. I completely agree with the general outline of your experimental approach. However, I miss information that would allow transferring your results to other locations. In its current form, the manuscript is merely an interesting case study. However, the importance of your work can be easily improved by adding a few extra data, such as:

Information on light intensities during the experiments. Average light intensity at noon, or the typical course of light intensity during a day would be sufficient. This would allow estimating how absorbed light energy is used (CO2 fixation, photorespiration, Asada cycle, etc.). The determination of the different fluorescence quenching types by PAM measurement are not sufficient to assess the positive effect of an electron sink that is mediated by nitrate reduction.

Response: Thank you for your suggestion. Thank you for decision and constructive comments on our manuscript. We had considered your suggestion carefully and made some changes. We have tried our best to improve and made some marks in the manuscript.

As a control you have used samples irrigated without added nitrogen fertilizer. But you did not provide information on the availability of ammonia or nitrate in the soil water phase of these controls.

What type and what quality of N-fertilizer did you use?

Response: Thank you for your suggestion. Thank you for decision and constructive comments on our manuscript. We had considered your suggestion carefully and made some changes. We have tried our best to improve and made some marks in the manuscript.

What was the ion-exchange capacity of the soil? -> Did you measure the concentration of relevant ions in the soil water phase during the course of your experiments?

Response: Thank you for decision and constructive comments on our manuscript. We had considered your suggestion carefully and made some changes. We have tried our best to improve and made some marks in the manuscript.

Finally: You have mentioned phosphate and potassium supply to the soil, but you do not provide any information on soil quality in general. Thus, your experiments can’t be reproduced elsewhere by colleagues. - Please improve.

Response: Thank you for your suggestion. We had considered your suggestion carefully and made some changes. We have tried our best to improve and made some marks in the manuscript.

Lay out:

Fig 1 → The figures are very small and therefore hard to read. Maybe they should be arranged in one column using page width.

Response: Thank you for decision and constructive comments on our manuscript. We had considered your suggestion carefully and made some changes Fig 1.

Figures should be self-explaining. Therefore, explain shortages in use in the figure legend, please.

Response: Thank you for decision and constructive comments on our manuscript. We had considered your suggestion carefully and made some changes Fig 1, Fig 2

Use of English language:

Obviously you have used a text interpreter. Nevertheless, the use of English language needs to be improved. Some words are not adequately translated. Therefore, it is hard to understand the meaning of some sentences (see the abstract, the conconclusion, for instrance).

Response: Thank you for your suggestion. Thank you for decision and constructive comments on our manuscript.

line 111 The term rate is not adequate, the meaning would be „change in nitrogen concentration per time interval“. You have applied an amount of nitrogen fertilizer per hectar.

Response: Thank you for decision and constructive comments on our manuscript. We had considered your suggestion carefully and made some changes L111.

Fig 2 →  The term „nitrogen rate“ is not adequate  (see line 111)   -  In Table 1 you correctly are using the term „N application“. Please make sure to use this term in your manuscript throughout.

Response: Thank you for decision and constructive comments on our manuscript. We had considered your suggestion carefully and made some changes Fig 2.

The conclusion chapter needs to be significantly improved. Sentences should be shorter, and all sentences need to contain a subject and a verb! 

content:

The complete discussion chapter is hard to read. Here not only the special use of English language is causing problems, but technical terms are not used correctly. SPAD, for instance, is a method to measure the concentration of photosynthetic active units per area. You are using this term to describe a chemical compound.

There is no clear line of arguments in the discussion subchapters, and some of the points you make are trivial.

Response: Thank you for your suggestion. Thank you for decision and constructive comments on our manuscript. We have tried our best to improve and made some marks in the manuscript.

The complete discussion chapter should be re-written in an understandable style.

Response: Thank you for your suggestion. Thank you for decision and constructive comments on our manuscript. We had considered your suggestion carefully and made some changes. We have tried our best to improve and made some marks in the manuscript.

Reviewer 3 Report

The work presented for review is relevant, it is very important to justify the use of nitrogen fertilizers. The section "Materials and Methods" is described very superficially. The abstract does not reflect the main results. This work is subject to complete revision and rewriting, each section.

English is difficult to understand

Author Response

Agronomy-2404439

Effects of Nitrogen Fertilizer on Photosynthetic and chlorophyll fluorescence Yield of Wheat (Triticum aestivum L.)

Dear Reviewer

Thank you very much for handling our manuscript. Thank you for your suggestion. Thank you for your decision and constructive comments on our manuscript. We had considered your suggestion carefully and made some changes. In the revised manuscript, we have attempted to address all the comments from the reviewers. All the changes that we have made in the revised manuscript are highlighted with track Changes in the track file. Below are our detailed point-to-point responses to the comments from the reviewers.

Thank you for your kind consideration.

Best regards,

Dr. Hafeez Noor

Key Laboratory of Functional Agriculture in the Loess Plateau, Ministry of Agriculture and Rural Affairs, Taigu,

College of Agronomy, Shanxi Agricultural University,

Taigu, Shanxi, 030801, P. R China

*********************************************************************

Review Report-3

The work presented for review is relevant, it is very important to justify the use of nitrogen fertilizers. The section "Materials and Methods" is described very superficially. The abstract does not reflect the main results. This work is subject to complete revision and rewriting, each section.

Response: Thank you for your suggestion. The manuscript has benefited from these insightful suggestions. Thank you for your decision and constructive comments on our manuscript. We had considered your suggestion carefully and made some changes. In the revised manuscript, we have attempted to address all the comments from the reviewers. Below are our detailed point-to-point responses to the comments from the reviewers.

Round 2

Reviewer 1 Report

Dear Editor

MDPI Agronomy

I am sharing my observations about the manuscript: “Nitrogen Fertilizer on Physiology Photosynthetic and chlorophyll fluorescence of Yield Food Nutrition Wheat (Triticum aestivum L.). The manuscript has the same problems. The authors didn’t answer the questions and also the results are as previous version. Tables showed significance for interaction but in the manuscript, the interactions weren’t written. The conclusion didn’t show what authors observed in the study. I don't feel qualified to judge about the English language and style, but there are paragraphs hard to understand. In my opinion is necessary the authors have more time for the corrections. 

Best regards.

12 May 2023.

Author Response

Agronomy-2404439

Effects of Nitrogen Fertilizer on Photosynthetic Characteristics and Yield

Dear Reviewer

Thank you very much for handling our manuscript. Thank you for your suggestion. Thank you for your decision and constructive comments on our manuscript. We had considered your suggestion carefully and made some changes. In the revised manuscript, we have attempted to address all the comments from the reviewers. All the changes that we have made in the revised manuscript. Below are our detailed point-to-point responses to the comments from the reviewers.

Thank you for your kind consideration.

Best regards,

Dr. Hafeez Noor

Key Laboratory of Functional Agriculture in the Loess Plateau, Ministry of Agriculture and Rural Affairs, Taigu,

College of Agronomy, Shanxi Agricultural University,

Taigu, Shanxi, 030801, P. R China

*********************************************************************

Review Report 1

I am sharing my observations about the manuscript: “Nitrogen Fertilizer on Physiology Photosynthetic and chlorophyll fluorescence of Yield Food Nutrition Wheat (Triticum aestivum L.). The manuscript has the same problems. The authors didn’t answer the questions and also the results are as previous version. Tables showed significance for interaction but in the manuscript, the interactions weren’t written. The conclusion didn’t show what authors observed in the study. I don't feel qualified to judge about the English language and style, but there are paragraphs hard to understand. In my opinion is necessary the authors have more time for the corrections.

Response: Thank you for your suggestion. The manuscript has benefited from these insightful suggestions. Thank you for your decision and constructive comments on our manuscript. We had considered your suggestion carefully and made some changes. In the revised manuscript, we have attempted to address all the comments from the reviewers. Below are our detailed point-to-point responses to the comments from the reviewers.

Reviewer 3 Report

Dear authors! you edited the article. She became much easier to understand. I ask you to improve it. Thank you. All sections need some work. your comments in the file.

material needs to improve English

Author Response

Agronomy-2404439

Effects of Nitrogen Fertilizer on Photosynthetic Characteristics and Yield

Dear Reviewer

Thank you very much for handling our manuscript. Thank you for your suggestion. Thank you for your decision and constructive comments on our manuscript. We had considered your suggestion carefully and made some changes. In the revised manuscript, we have attempted to address all the comments from the reviewers. All the changes that we have made in the revised manuscript. Below are our detailed point-to-point responses to the comments from the reviewers.

Thank you for your kind consideration.

Best regards,

Dr. Hafeez Noor

Key Laboratory of Functional Agriculture in the Loess Plateau, Ministry of Agriculture and Rural Affairs, Taigu,

College of Agronomy, Shanxi Agricultural University,

Taigu, Shanxi, 030801, P. R China

*********************************************************************

Review Report-3

Dear authors! You edited the article. She became much easier to understand. I ask you to improve it. Thank you. All sections need some work. Your comments in the file.

Response: Thank you for your suggestion. The manuscript has benefited from these insightful suggestions. Thank you for your decision and constructive comments on our manuscript. We had considered your suggestion carefully and made some changes. In the revised manuscript, we have attempted to address all the comments from the reviewers. Below are our detailed point-to-point responses to the comments from the reviewers.
